# Molecular Dynamics Simulation Analysis of JAK1 Initial Activation: Phosphorylation-Induced Conformational Dynamics and Domain Interactions

**DOI:** 10.3390/life15081316

**Published:** 2025-08-19

**Authors:** Xinyu Peng, Kefu Liu, Guodong Chen, Shengjie Sun

**Affiliations:** 1Department of Biomedical Informatic, School of Life Sciences, Central South University, Changsha 410083, China; pengxy07@163.com (X.P.); liukefu@csu.edu.cn (K.L.); 2Center for Medical Genetics, School of Life Sciences, Central South University, Changsha 410083, China

**Keywords:** JAK, JAK activation, electrostatic, Delphi, tyrosine phosphorylation, molecular dynamic simulations

## Abstract

Janus kinase is critical for cytokine-mediated signaling, and its hyperactivation due to mutations drives various diseases. The activation of Janus kinase 1 (JAK1) involves a conformational transition from a closed to an open state, but the underlying mechanism remains unclear. This study investigates the roles of two tyrosine residues, Y1034 and Y1035, within the activation loop of the tyrosine kinase domain. Molecular dynamics simulations reveal that phosphorylation, particularly bisphosphorylation at Y1034 and Y1035, promotes the transition to the open conformation, with pY1035 exerting a greater influence than pY1034. Phosphorylation increases the negative charge on the TK domain surface, facilitating its dissociation from the FERM domain, while also weakening TK-FERM interactions. However, the loop between the TK and PK domains formed stable hydrogen bonds with other domains, hindering the full activation process. Using 1 µs molecular dynamics simulations is not sufficient for full activation. These findings elucidate the molecular mechanisms governing the JAK1 initial activation and provide insights for targeting its regulation in disease contexts.

## 1. Introduction

Janus kinase (JAK) is a kind of intracellular receptor tyrosine kinase, including JAK1, JAK2, JAK3 and Tyrosine kinase 2 (TYK2) [1,2]. JAK and its downstream Signal Transducers and Activators of Transcription (STAT) constitute the JAK/STAT signaling pathway [3]. When cytokines bind to the extracellular domain of their specific receptors, they cause the receptor dimerization. The dimerization leads to JAK autophosphorylation and activation [4,5,6], subsequently activating the downstream signaling pathway to regulate specific gene transcription [7]. The JAK/STAT pathway is an evolutionarily conservative signaling pathway in organisms that plays a role in many important physiological processes. Deregulation of JAK/STAT signaling is often associated with the pathogenesis of autoimmune diseases and hematopoietic malignancies [8]. Different types of leukemia have genomic aberrations in all four JAKs, most of which are activating somatic mutations [9,10,11]. Further understanding of the mechanism of JAK activation will help to better study the development of these diseases.

JAKs contain four distinct conserved domains: an N-terminal Four-point-one protein, Ezrin, Radixin, Moesin (FERM) domain; an Src Homology 2 (SH2) domain; a Pseudo-Kinase (PK) domain and a Tyrosine Kinase (TK) domain [12]. TK is regarded as the domain with catalytic ability, and PK generally lacks catalytic activity but is highly significant for regulating the catalytic activity of TK [13,14,15]. The activation segment influences both protein-substrate binding and catalytic efficiency [16,17]. This segment in Janus kinases contains two adjacent conserved phosphorylatable tyrosines [18]. Phosphorylation of these two tyrosine residues (Tyr) was found to play an important role in the activation of JAK [19,20,21,22]. Specifically, JAK1 and JAK3 are controlled through distinct patterns of tyrosine phosphorylation [19]. For JAK2 to become catalytically active, phosphorylation of Y1007 within its kinase activation loop is essential. Similarly, the activation of TYK2 in response to interferon-alpha depends on the phosphorylation of specific positive regulatory tyrosine by a separate kinase.

The insulin receptor TK domain activation segment also has two adjacent tyrosine sites; the structure showed that these tyrosines in the activation segment block the ATP binding site, and phosphorylation helps open the region [23,24]. It implies that the phosphorylation status of the two Tyr is important in JAK structural rearrangement from a compact “closed” state to an extended “open” state [25,26]. But the influence of conformation by this phosphorylation status is still unknown.

Recently Glassman et al. [25] reported the 3.6 Å resolution cryo-EM structure of mouse full-length JAK1 complexed with a cytokine receptor intracellular Box1/Box2 domain. Based on the development of relevant algorithms and advanced software, we can perform molecular dynamics simulations to quantify the effects of phosphorylation on the structure of JAKs to provide valuable insights into the mechanism of JAK activation. In order to explore whether the conformational change during JAK activation is caused by phosphorylation, we built an autoinhibition model of full-length JAK1 based on the previously reported structure of full-length JAK1 and the structure of the autoinhibitory TYK2 PK-TK domain fragment to investigate the effect of phosphorylation on the conformation of JAK1 through molecular dynamics simulations. The results showed that the phosphorylation of Y1034 and Y1035 could change the conformation of JAK1 to some extent, and the TK domain and FERM domain tended to be separated in the spatial structure. Furthermore, phosphorylation affected the distribution of surface charge, increasing the negative charge of the TK domain, which may more easily separate from the FERM domain. Additionally, the effect of hydrogen bonds between the TK domain and the FERM domain was weakened after phosphorylation.

## 2. Method

### 2.1. JAK1 Modeling

The autoinhibited full-length JAK1 structure was built through structure alignment of the dimerized activated full-length JAK1 structure (PDB ID: 7T6F [25]) and autoinhibited PK-TK dimer (PDB ID: 4OLI [15]). We used ChimeraX 1.7 (UCSF, San Francisco, CA, USA) aligned with the PK domains of full-length activated JAK1 (FERM-SH2-PK-TK) and autoinhibited JAK1 (PK-TK). Subsequently, we removed the tyrosine kinase (TK) domain from the inhibited JAK1 and the PK domain from the activated JAK1 to construct the autoinhibited full-length JAK1 structure [27,28]. Targeting two conserved tyrosine residues Y1034 and Y1035 within the known activation segment of the JAK1 TK domain, three phosphorylated protein models (pY1034 JAK1, pY1035 JAK1, and pY1034pY1035 JAK1) were built using CHARMM-GUI (Lehigh University, Bethlehem, PA, USA) [29], which were used for further molecular dynamics simulations to investigate the effects of pY1034 and pY1035 on the conformation of JAK1.

### 2.2. Molecular Dynamics Simulation

The CHARMM-GUI webserver was used to create the simulation system. The protein was solvated in a cubic water box (TIP3P [30]) with a minimum distance of 15 Å between the protein surface and the box boundaries. NaCl was used to ionize the system with a concentration of 150 mM. Parameterization of the atoms in the system was performed using the CHARMM36 force field [31]. Periodic boundary conditions were applied to the simulation box, and particle mesh Ewald [32] was used for the long-range electrostatic interactions. The final system was then subjected to molecular dynamics simulation using NAMD3.0 [33]. The whole simulation included two steps. The first was equilibration, and the second was the production run. NVT was used for the first step. The temperature was set at 310.15 K, using a Langevin thermostat with a damping coefficient of 1/ps. The pressure was set to 1 atm, using a Nosé–Hoover Langevin piston barostat with a decay period of 25 fs. The temperature was reassigned every 500 steps. During the equilibration, a constraint was applied to the protein. In the production run, NPT was run continuously for 100 ns and the constraint on the protein was released. For nonbonding interactions (electrostatic and van der Waals), the cutoff was set to 12 Å. The switching distance was set to 10 Å. Simulations were performed for each of the four protein models constructed. The production run in pY1034pY1035 JAK1 system was repeated and extended to a 1 µs MD simulation.

### 2.3. Root Mean Square Deviation (RMSD) and Root Mean Square Fluctuation (RMSF)

RMSD is an important metric used to measure the average distance between two protein structures to gauge the degree of conformational difference or trajectory stabilization, calculated using Equation (1).(1)$$\mathrm{RMSD}=\sqrt{\frac{{\sum_{i=1}^{N}\left[w_{i}\left(r_{i}-r_{i}^{ref}\right)\right]}^{2}}{WN}}$$
where W = Σw_i_ is the weighting factor, N is the total number of atoms, r_i_ (t) is the position of atom i at time t after least squares fitting the structure to the reference structure, and r_i_^ref^ is the reference position of residue i defined by the reference structure (here we used the initial structure as the reference).

RMSF is used to calculate the magnitude of change of each atom with respect to its average position and characterizes the flexibility of the molecular structure. The RMSF of Cα for residues (32-1153) of JAK1 was calculated based on the steady state (last 10 ns) using Visual Molecular Dynamics (VMD1.9.3) [34], with Equation (2).(2)$${\mathrm{RMSF}}_{i}=\sqrt{\frac{\sum_{t_{j}=1}^{T}\left[r_{i}(t_{j})-r_{i}^{ref}\right]}{T}}$$
where i represents the residue ID, T represents the total simulation time (here, the number of frames), r_i_ (t_j_) represents the position of residue i at the time of t_j_, and r_i_^ref^ is the reference position of the residue i, calculated from the time-average position.

### 2.4. Hydrogen Bonding

The hydrogen bond formation between the TK domain and the FERM domain was analyzed using VMD for the stable segment (the last 10 ns). The cutoff distance and the angle for the hydrogen bonds were set as 3.0 Å and 20°.

### 2.5. Electrostatic Potential Study

The structures of phosphorylated and non-phosphorylated JAK1 used to calculate the electrostatic potential were from the last frame (at 100 ns) of the simulations. The electrostatic potential calculations of the four models were carried out using Delphi 8.5 (Clemson, SC, USA) [35]. The charge and the radius of atoms were calculated using the force field CHARMM36 and assigned using PDB2PQR 3.7.1 (St. Louis, MO, USA) [36]. The dielectric constants were set as 2 for proteins, and 80 for water. The salt concentration was set as 150 mM, and the probe radius was set as 1.4 Å. The protein filling ratio was set as 70%, and the resolution was set as 1.5 grids/Å. The electrostatic potential on the surface of molecules was visualized by Chimera with the color range set from −1.0 kT/e (red) to 1.0 kT/e (blue).

### 2.6. Comparison of Conformational Differences Between Phosphorylated and Non-Phosphorylated JAK1

The phosphorylated and non-phosphorylated JAK1 forms were compared to show the conformational differences. The structures were from the last frame (at 100 ns) of the simulations. The mass centers of FERM, PK and TK in the four models (last 10 ns) were calculated, to compare the distance between the TK domain and the FERM domain, as well as the angle of FERM-PK-TK. The opening angle was calculated by the mass centers of the FERM, SH2, and TK domains.

## 3. Results

### 3.1. Tyr Phosphorylation in Activation Segment Has an Effect on the Conformation of JAK1

To investigate the effect of Tyr phosphorylation in the activation segment on the conformation of JAK1 in its autoinhibited state, the following steps were taken. The autoinhibited full-length JAK1 structure was built. Then, phosphate groups were added to form the four models (nonp JAK1, pY1034 JAK1, pY1035 JAK1, and pY1034pY1035 JAK1). They were subjected to 100 ns molecular dynamics simulations. The RMSD result showed that the conformation variation was relatively stable after 60 ns (Appendix A). So, the last 10 ns of the simulation result was used in subsequent analysis. To investigate the effect of phosphorylation on the induction of structural rearrangements of JAK1, we compared the molecular structures of phosphorylated and non-phosphorylated JAK1 molecules. We observed that there was no great difference in the global conformation of the four molecules (Figure 1).

However, the gap between the opening loop of TK and the FERM domain has changed. pY1034pY1035 JAK1 was seen to have the largest gap and nonp JAK1 has the smallest gap. By quantifying the distance between the TK domain and the FERM domain, it was observed that there was an increase in the distance between the mass centers between TK and FERM of pY1035 JAK1 and pY1034pY1035 JAK1 compared to nonp JAK1 and pY1034 JAK1 (Figure 2b). In addition, the opening angle size of JAK1 nonp JAK1 < pY1034 JAK1 < pY1035 JAK1 < pY1034pY1035 JAK1 (Figure 2c). In summary, the results show that phosphorylation still has a large effect on interaction between the FERM and TK domains of JAK1, in which the effect of bisphosphorylation is larger than that of monophosphorylation, and the effect of pY1035 is larger than that of pY1034.

### 3.2. Surface Negative Charge of TK Domain Increased After Phosphorylation in Y1034/Y1035

Since pY1034 and pY1035 phosphorylation alters the interaction between the TK and FERM domains, the following effects were observed. The variation in surface negative charge may affect the intermolecular forces to alter the interaction distance. The surface electrostatic potential of JAK1 in the phosphorylated and unphosphorylated models was calculated. The surface of electrostatic potential of the adjacent regions between the TK domain and FERM domain was observed. The charge distribution of the adjacent regions in the TK domain changed, where the adjacent regions in the FERM domain did not change to a large extent (Figure 3). This indicates that the phosphorylation state changes the negative charge distribution in the domain in which they were located. The negatively charged area (NCA) in the TK adjacent regions is greatest in pY1034pY1035 JAK1. The NCA in pY1035 JAK1 is greater than pY1034 JAK1 in TK adjacent regions. The FERM adjacent regions also have a large NCA. Thus, there was the potential for greater repulsion between the two domains in the pY1034pY1035 JAK1 model, which may drive JAK1 to an open structure. In summary, phosphorylation may alter the distribution of existing charges, thus affecting the charge distribution in the closed regions of the TK and FERM domains, to induce conformational changes. The change in electrostatic potential is greater in the bisphosphorylated state than in the monophosphorylated state, and pY1035 had a greater effect on the change in electrostatic potential than pY1034.

### 3.3. Phosphorylation in Y1034/Y1035 Leads to Weakening of Hydrogen Bonding Between TK and FERM

As shown in the previous conformational comparison and electrostatic analysis, phosphorylation, especially bisphosphorylation, promoted the separation between TK and FERM. We therefore hypothesize that the loss of hydrogen bonding between TK and FERM may lead to an easier separation in the two domains and thus contribute to the activation of JAK1.

Analyzing the high occupancy hydrogen bonds (occupancy rate ≥ 50%) of the four protein models revealed that nonp JAK1 (Figure 4a) and pY1034 JAK1 (Figure 4b) had three high occupancy hydrogen bonds. But pY1035 JAK1 (Figure 4c) and pY1034pY1035 JAK1 (Figure 4d) had only one high occupancy hydrogen bond. Particularly, the highest hydrogen bond occupancy rate reached only 50.00% in the pY1034pY1035 JAK1 model (Figure 4d). The decrease in the number of high occupancy hydrogen bonds is likely to be an important factor in promoting the separation between the TK domain and the FERM domain. In addition, we could also see that the high occupancy hydrogen bonds of nonp JAK1 were mainly concentrated at the outer part of the protein opening, suggesting that their presence makes the outer part of the nonp JAK1 opening more stable and less prone to the separation between TK and FERM. On the other hand, the high-occupancy hydrogen bonds of pY1034 JAK1, pY1035 JAK1 and pY1034pY1035 JAK1 tended to be more inwardly oriented, suggesting that the openings of these proteins are relatively less stable, and the openings are more prone to segregation from the outside. Filtering the hydrogen bonds with occupancy greater than 10.00% from all the predicted hydrogen bonds, it can be seen that pY1034pY1035 JAK1 (Figure 4h) has a smaller number of hydrogen bonds than nonp JAK1 (Figure 4e), pY1034 JAK1 (Figure 4f), and pY1035 JAK1 (Figure 4g), and also has a significantly smaller occupancy than most of the hydrogen bonds of the latter three. In addition, the number of hydrogen bonds in pY1035 JAK1 is significantly lower than in pY1034 JAK1.

Taken together, these results suggest that phosphorylation leads to weakening of the hydrogen bonding between the TK domain and the FERM domain, which promotes the conformational change in the previous section. Furthermore, the weakening of the hydrogen bonding between TK and FERM after bisphosphorylation was the most significant. Additionally, compared to pY1034, the weakening of the hydrogen bonding between TK and FERM after pY1035 was more significant.

### 3.4. Phosphorylation in Y1034 and Y1035 Changed JAK1 Structural Stability in Partial Amino Residue

In order to explore the relationship between the flexibility of individual regions and the conformational changes of JAK1 by phosphorylation, we performed RMSF calculations based on the phosphorylated and unphosphorylated JAK1 in the steady state, respectively. We found that the fluctuation trends of individual residues of four models are approximately the same, but differences in the fluctuation of some regions in the phosphorylated JAK1 compared to the unphosphorylated JAK1 could be observed. Among the four proteins, the regions with obvious differences in RMSF values were located in the FERM domain (Appendix A), where the RMSF values of phosphorylated JAK1 were higher than those of nonp JAK1, with those of pY1035 JAK1 being higher than those of pY1034 JAK1 and pY1034pY1035 JAK1. Because it is located at the end of the FERM domain, we hypothesize that it may be related to the separation of the FERM domain from the TK domain. (2) They were located within the SH2 domain (Appendix A), and here the RMSF values of pY1034pY1035 JAK1 are slightly higher than those of the others. Since the SH2 domain is mainly involved in stabilizing the structural conformation of JAK kinase [37], it may be related to the fact that bisphosphorylation is more likely to cause conformational changes. (3) They were located within the PK domain (Appendix A), where the RMSF values of unphosphorylated JAK1 are significantly higher than those of phosphorylated JAK1. The PK domain is critical for regulating the catalytic activity of TK, and it was found that the PK domain has an autoinhibitory effect [13,14,15]. Thus, we hypothesized that the decrease in the flexibility of this region may attenuate this autoinhibitory effect thereby leading to a conformational shift from an inhibited closed state to an open state after phosphorylation.

### 3.5. MD Simulation Lasting for 1.0 µs Further Provides the Stuck Activation of JAK1 from the Autoinhibited State

In the multi-phosphorylation studies, pY1034pY1035 JAK1 exhibited the fewest hydrogen bonds between the tyrosine kinase (TK) and FERM domains. To investigate the activation process, we extended the molecular dynamics (MD) simulation to 1 µs with duplicate runs. However, the system remained far from full activation, showing only a slight opening (Figure 5d). A 1 µs molecular dynamics (MD) simulation may be insufficient to capture the full activation process of pY1034pY1035 JAK1. Although the RMSD with frame 1 as the reference (red) stabilizes after 600 ns, the RMSD with frame 1000 as the reference (blue) continues to decrease (Figure 5a). This suggests that the overall structure remains dynamic and continues to fluctuate. To validate our hypothesis, we calculated the Root Mean Square Fluctuation (RMSF) of pY1034pY1035 JAK1 for distinct time intervals, depicted in different colors and patterns (Figure 5b). The FERM domain exhibits a higher RMSF during the final 100 ns of the MD simulation, whereas most residues in the TK domain show lower RMSF. Notably, specific residues, including I1060–W1065 and T1100–V1105, display pronounced fluctuations in the later periods. Additionally, we computed the average RMSF of all Cα atoms across different time intervals, revealing a sharp decline followed by a slight increase (Appendix A).

In this 1 µs MD simulation study of pY1034pY1035 JAK1, we expanded our hydrogen bond analysis to include interactions between the TK domain, including its loop region (residues 856–874, previously excluded from the 100 ns MD simulation analysis) connecting to the PK domain, and all other domains, beyond solely the FERM domain, to identify critical residue pairs stabilizing the autoinhibited conformation. We identified the LYS328–GLU927 hydrogen bond with over 60% occupancy in the 1 µs simulation. Additionally, ASN297 and SER295 consistently formed hydrogen bonds with GLU858, located in the TK-PK loop (residues 856–874). Notably, nearly all stable hydrogen bonds were localized in this region (Figure 5c,d), suggesting that the loop between TK and PK plays a critical role in maintaining the autoinhibited state of JAK1.

## 4. Discussion

This study elucidates the role of phosphorylation at tyrosine residues Y1034 and Y1035 in the conformational dynamics of JAK1 activation. Our molecular dynamics simulations reveal that phosphorylation, particularly bisphosphorylation at Y1034 and Y1035, promotes a conformational shift from a closed to an open state, primarily by increasing the spatial separation between the TK and FERM domains. This is driven by an enhanced negative charge on the TK domain surface and weakened hydrogen bonding between the TK and FERM domains. Notably, pY1035 exerts a stronger influence than pY1034 on these structural changes, suggesting a hierarchical role of these residues in JAK1 activation.

Previous work by Liu et al. demonstrated that mutating Y1034 to phenylalanine abolished autophosphorylation and STAT5A activation, while mutating Y1035 (JAK1YF mutant) permitted residual phosphorylation and activation. These findings suggest that Y1034 is critical for catalytic activity, whereas Y1035 may be dispensable for catalysis but pivotal for conformational rearrangement. Our results align with this, indicating that pY1035 significantly enhances the open conformation, potentially facilitating access to the ATP-binding site, while pY1034 may primarily support catalytic function. This distinction underscores a nuanced interplay between phosphorylation sites in balancing structural and catalytic roles during JAK1 activation.

The observed increase in negative charge on the TK domain following phosphorylation likely enhances electrostatic repulsion with the FERM domain, promoting domain separation. Concurrently, the reduction in high-occupancy hydrogen bonds, especially in the bisphosphorylated state, destabilizes the closed conformation, further enabling the transition to an open state. The RMSF analysis highlights regional flexibility changes, particularly in the FERM, SH2, and PK domains, which may modulate autoinhibition and stabilize the open conformation. Notably, reduced flexibility in the PK domain post-phosphorylation may attenuate its autoinhibitory effect, facilitating TK activation. A new reported JAK1 mutation S404P, which near high-occupancy hydrogen bonds in the nonp JAK1 model, enhanced cancer cell invasion ability [38]. A new case of chronic eosinophilic leukemia is sensitive to JAK1 inhibitor treatment and has a JAK1 R629D630del mutation, located in the flexibility changed PK region [39]. K1026E and Y1035C, which have high-occupancy hydrogen bonds, were identified by IL-3 independent clone screening and provide evidence that they influence JAK1 phosphorylation [40]. These findings suggest that mutations in these regions may influence conformational changes and Y1034/Y1035 phosphorylation to activate the JAK1 signaling pathway.

Unexpectedly, we found that the TK-PK loop, a junction linker between TK and PK, seems have a strong repressive effect in JAK1 structural transformation. Most mutations are found in four domains, and the linker effect in structure and function has rarely been investigated. One study found that palmitoylated C541 and C542 in the JAK1 SH2-PK linker was important for Y1034/Y1035 phosphorylation and activation [41] which suggested that the junction sequence between two domains may also have important roles in structure rearrangement and enzyme activation.

Despite these insights, limitations remain. Our simulations focus on Y1034 and Y1035, but other phosphorylation sites within JAK1 or other JAK family members may also influence activation. Additionally, the precise link between conformational changes and catalytic activation requires further experimental validation, such as kinase activity assays or structural studies of phosphorylated JAK1 in complexes with substrates. The role of the TK-PK loop in JAK1 needs further investigation.

## 5. Conclusions

In the autoinhibited state, JAK1’s FERM and TK domains are stabilized by numerous high-occupancy hydrogen bonds. Our electrostatic and conformational analyses reveal that bisphosphorylation at Y1034 and Y1035 promotes a greater angle and increased distance between FERM and TK (nonp JAK1 < pY1034 JAK1 < pY1035 JAK1 < pY1034pY1035 JAK1), driven by an enhanced negative charge on the TK domain that favors an open conformation. Additionally, phosphorylation reduces both the number and overall occupancy of high-occupancy hydrogen bonds, particularly near the outer part of the opening site, further facilitating the structural transition. However, the 1 µs molecular dynamics simulation for bisphosphorylation did not capture the full activation process, likely due to persistent hydrogen bonds between the TK-PK loop and other domains that hinder complete opening. Further analysis is needed to elucidate the full activation mechanism.

## Figures and Tables

**Figure 1 life-15-01316-f001:**
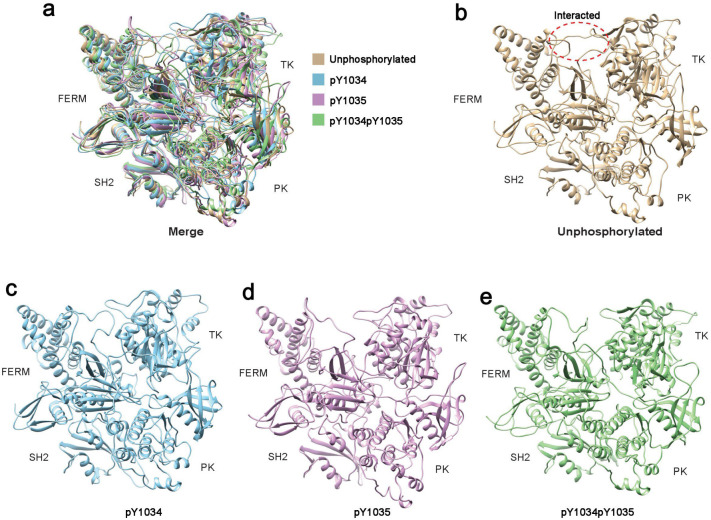
Conformational comparison of unphosphorylated and phosphorylated JAK1. (**a**) Comparison of overlapping conformations of the four models. (**b**–**e**) Conformations of each of the four models in (**a**), respectively.

**Figure 2 life-15-01316-f002:**
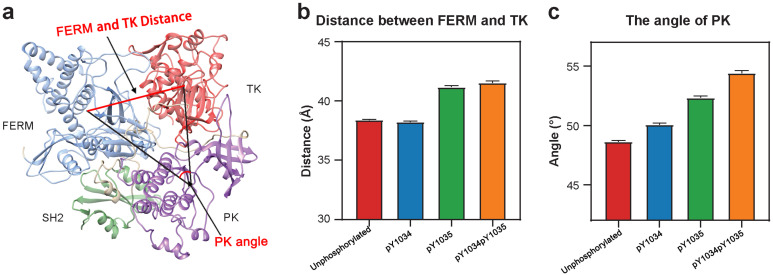
Quantitative comparison of unphosphorylated and phosphorylated JAK1 structures. (**a**) Schematic of the mass centers of FERM, PK and TK (**left**) and calculated side and angle (**right**). (**b**) The distance of FT. (**c**) The angle of ∠α.

**Figure 3 life-15-01316-f003:**
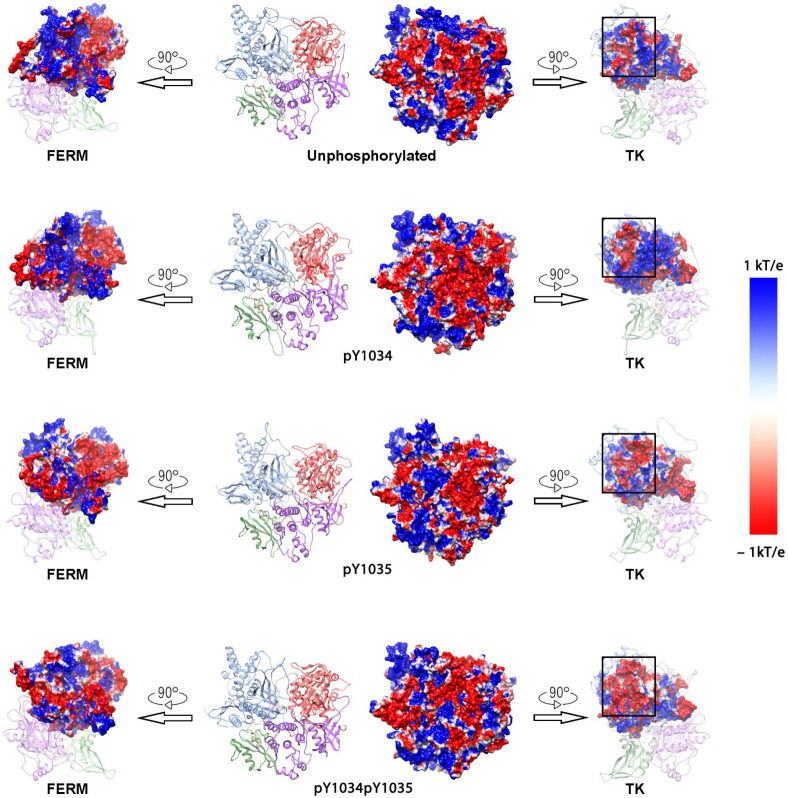
Electrostatic surface of unphosphorylated and phosphorylated JAK1. The surface electrostatic surface of the FERM domain is represented in the first column, and the electrostatic surface of the TK domain is represented in the last column. The color range was set from −1.0 kT/e (red) to 1.0 kT/e (blue).

**Figure 4 life-15-01316-f004:**
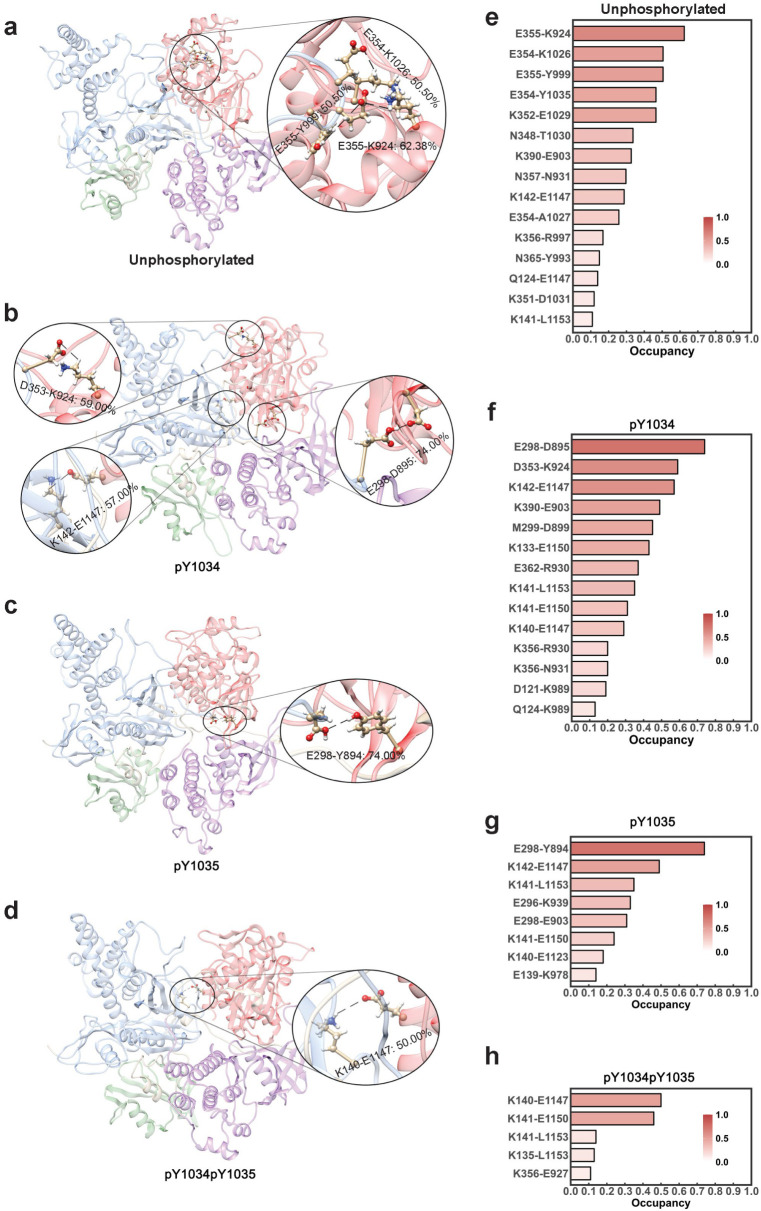
Hydrogen bonds formed between the FERM domain and the TK domain: (**a**–**d**) are hydrogen bonds with high occupancy of each of the four proteins, respectively; (**e**–**h**) are the occupancy of hydrogen bonds with occupancy greater than 10% of each of the four proteins, respectively.

**Figure 5 life-15-01316-f005:**
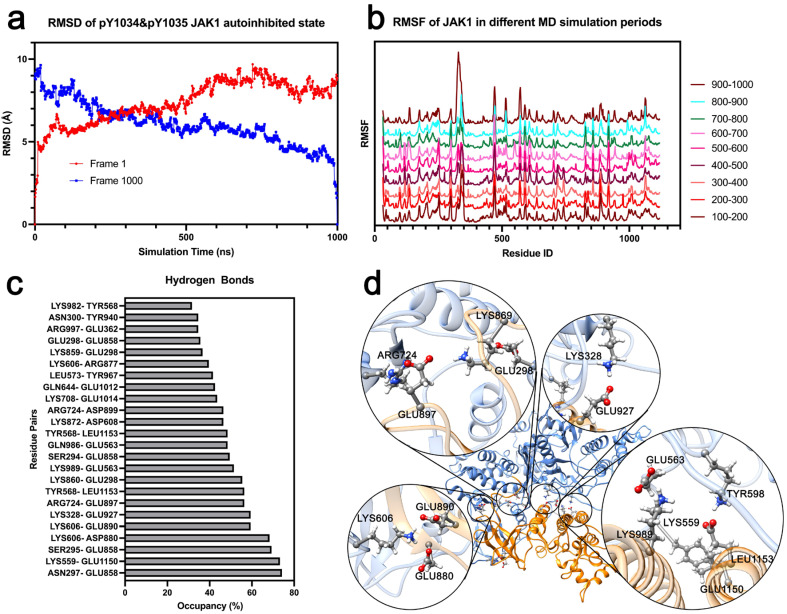
Molecular Dynamics (MD) Simulation Analysis of pY1034pY1035 JAK1 over 1 µs. (**a**) Root Mean Square Deviation (RMSD) of JAK1-pY1034pY1035 over 1000 ns, with reference frames at 0 ns (red) and 1000 ns (blue). (**b**) Root Mean Square Fluctuation (RMSF) across different time intervals (100–200 ns, 200–300 ns, …, 900–1000 ns). (**c**) Hydrogen bond analysis between TK domain and other domains in JAK1 during the final 100 ns of the MD simulation. (**d**) High-occupancy hydrogen bonds mapped onto the JAK1 structure.

## Data Availability

The original contributions presented in this study are included in the article and Appendix A. Further inquiries can be directed to the corresponding authors.

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
