# Peer review of "Molecular Dynamics Simulation Analysis of JAK1 Initial Activation: Phosphorylation-Induced Conformational Dynamics and Domain Interactions"

_life, 2025, doi:10.3390/life15081316_

Round 1
Reviewer 1 Report (Previous Reviewer 1)
Comments and Suggestions for Authors
The manuscript has improved significantly, and I appreciate the authors’ revisions. However, a few details still require clarification and elaboration:
1- Please specify the software used for the sequence alignment and the removal of unwanted regions during the JAK1 modeling process.
2 - The methods section should include a detailed description of the molecular dynamics (MD) steps, particularly for both the initial 100 ns simulations and the 1 μs run. Please explain the rationale also.
3 - Clarify how the mass centers of the FERM, PK, and TK domains were defined. Additionally, provide a clear explanation of how the opening angle was calculated.
4 - There appears to be a discrepancy in the reported hydrogen bond interactions for pY1034/pY1035 JAK1 between the 100 ns and 1 μs simulations. In the 100 ns run, interactions include K140–E1147 and K141–E1150, while the 1 μs simulation identifies LYS328–GLU927 as the dominant interaction. Moreover, Figure 5C does not show any interactions involving residues K135–141. The authors should explain the reason for this shift in residue pairing and provide a clear interpretation of these results in the context of the structural dynamics.
Author Response
The manuscript has improved significantly, and I appreciate the authors’ revisions. However, a few details still require clarification and elaboration:
1- Please specify the software used for the sequence alignment and the removal of unwanted regions during the JAK1 modeling process.
Response: Thanks very much for this comment. We added the details in how to design protein structure modeling by ChimeraX. It has been highlighted in the method section 2.1.
2- The methods section should include a detailed description of the molecular dynamics (MD) steps, particularly for both the initial 100 ns simulations and the 1 μs run. Please explain the rationale also.
Response: Thanks very much for this comment. There is no significant change in modelling and MD configuration for JAK1pY1034/pY1035 1 us MD simulation. We added and highlighted the details in MD simulation method part (section 2.2).
3- Clarify how the mass centers of the FERM, PK, and TK domains were defined. Additionally, provide a clear explanation of how the opening angle was calculated.
Response: Thanks very much for this comment. The opening angle was calculated by the mass centers of the FERM, SH2, and TK do-mains. We added and highlighted it in the conformation difference part (section 2.6)
4 - There appears to be a discrepancy in the reported hydrogen bond interactions for pY1034/pY1035 JAK1 between the 100 ns and 1 μs simulations. In the 100 ns run, interactions include K140–E1147 and K141–E1150, while the 1 μs simulation identifies LYS328–GLU927 as the dominant interaction. Moreover, Figure 5C does not show any interactions involving residues K135–141. The authors should explain the reason for this shift in residue pairing and provide a clear interpretation of these results in the context of the structural dynamics.
Response: Thanks very much for this comment. We anticipated that the 1 μs MD simulations of the JAK1 pY1034/pY1035 system would reveal the activation process, but it did not yield the expected results. To understand the cause of this failure, we expanded our hydrogen bond analysis to encompass interactions between the tyrosine kinase domain, including its loop region (residues 856–874, previously excluded from the 100 ns MD simulation analysis), the pseudokinase domain, and all other domains, beyond just the FERM domain. This comprehensive approach aimed to identify key residue pairs stabilizing the autoinhibited conformation. The explanation was added and highlighted in the result sections (section 3.5).
Reviewer 2 Report (New Reviewer)
Comments and Suggestions for Authors
Activation of kinases is related to their interdomain interactions, and triggered by signaling-related post-translational modifications (PTMs), including tyrosine phosphorylation. In this manuscript by Peng et al, the authors have investigated the effects of mono- and bis-phosphorylation on tyrosine residues in the JAK1 kinase using molecular dynamics simulations, and learned about differences in the conformations that are sampled as a result of these PTMs.
Overall, I found this to be an interesting investigation, and the presentation of results and figures is very good. I have a few suggestions to improve the readability of this article and the thoroughness of analysis.
Specific critiques:
- It was not clear why portions of the manuscript were highlighted, this was quite distracting.
- The title in the manuscript and supplementary documents do not match the submission title.
- In the Abstract, the highlighted sentences (lines 21-25) seem redundant with sentences that appear later on (lines 35-39).
- Line 25-26 seems like it should be the 2nd sentence of the Abstract, as it defines JAK1 acronym.
- Line 109, unclear what the ‘thickness 15 Å’ refers to.
- Figure 1, missing (a) and (b) labels. Also, would be beneficial to the reader if all panels had the domains labeled like the overlaid structure does.
- What is the location on the structure of the two phosphorylated tyrosine residues? Could you show a figure panel with this region magnified?
- Figure 2, is the angle between FERM-PK-TK? Instead of a closed triangle on the structure, could be an open V-shape with the angle indicated on figure?
- Out of curiosity, are any new hydrogen bonds or other interactions developed with the phosphorylated tyrosine residues?
- Line 284, subheading reads as “3.5.1 µs” instead of “3.5 1 µs”, maybe say 1.0 µs so it is clear.
- For such a large multi-domain structure, the RMSD/RSMF will be affected by any interdomain re-arrangements. I would suggest doing the same analyses on the individual domains as well, i.e. isolating the FERM domain or TK domain and seeing how they change, or perhaps the two together as well. This would reveal the magnitude of fluctuations specific to the regions of interest.
Mostly, the English is acceptable. There are some spots with numerous spelling and grammatical issues that should be improved, but the meaning is still interpretable regardless. Ideally, these can be fixed or addressed by editor.
Author Response
Activation of kinases is related to their interdomain interactions, and triggered by signaling-related post-translational modifications (PTMs), including tyrosine phosphorylation. In this manuscript by Peng et al, the authors have investigated the effects of mono- and bis-phosphorylation on tyrosine residues in the JAK1 kinase using molecular dynamics simulations, and learned about differences in the conformations that are sampled as a result of these PTMs.
Overall, I found this to be an interesting investigation, and the presentation of results and figures is very good. I have a few suggestions to improve the readability of this article and the thoroughness of analysis.
Specific critiques:
- It was not clear why portions of the manuscript were highlighted; this was quite distracting.
Response: We are so sorry for the confusion caused by the highlighted portions in the manuscript. It is a re-submission manuscript. These highlights were intended to clearly indicate the revisions made in response to the previous comments. We recognize that this formatting may have been distracting, and we regret any inconvenience this may have caused.
- The title in the manuscript and supplementary documents do not match the submission title.
Response: We sincerely apologize for the error. The manuscript title is now finalized. We have updated it in the supplementary documents.
- In the Abstract, the highlighted sentences (lines 21-25) seem redundant with sentences that appear later on (lines 35-39).
Response: We sincerely apologize for the error. We have revised the abstract.
- Line 25-26 seems like it should be the 2nd sentence of the Abstract, as it defines JAK1 acronym.
Response: We are so sorry for the mistaken. We have revised it and make acronym appear at first time.
- Line 109, unclear what the ‘thickness 15 Å’ refers to.
Response: Thank you. The term "thickness 15 Å" refers to the distance of the water layer (solvation box) extending beyond the protein surface in all directions, as implemented in the molecular dynamics (MD) simulation setup to ensure adequate solvation of the JAK1 pY1034/pY1035 system. To improve clarity, we have revised Line 109 in the manuscript to explicitly state: "The protein was solvated in a cubic water box with a minimum distance of 15 Å between the protein surface and the box boundaries." We have also updated the materials with a detailed description of the solvation parameters for transparency (section 2.2).
- Figure 1, missing (a) and (b) labels. Also, would be beneficial to the reader if all panels had the domains labeled like the overlaid structure does.
Response: Thanks for your advice. We have add labeled in all structure in Fig. 1.
- What is the location on the structure of the two phosphorylated tyrosine residues? Could you show a figure panel with this region magnified?
Response: Thank you for your advice. To clarify, the two phosphorylated tyrosine residues (pY1034 and pY1035) in the JAK1 tyrosine kinase domain are highlighted within the dashed circle in Figure 1 to indicate their critical role in the activation loop.
- Figure 2, is the angle between FERM-PK-TK? Instead of a closed triangle on the structure, could be an open V-shape with the angle indicated on figure?
Response: Thanks for your advice. We have added label to indicated the angle.
- Out of curiosity, are any new hydrogen bonds or other interactions developed with the phosphorylated tyrosine residues?
Response: Thank you for your question. Phosphorylation negatively charges the JAK1 TK domain, repelling it from the FERM domain. Thus, no hydrogen bonds or salt bridges connect the FERM domain to other domains in the autoinhibited state. We will investigate potential interactions post-activation.
- Line 284, subheading reads as “3.5.1 µs” instead of “3.5 1 µs”, maybe say 1.0 µs so it is clear.
Response: Thanks for your advice. We have revised the subheading to “MD simulation last for 1.0 µs further provides the stuck the activation of JAK1 from autoinhibited state.”
- For such a large multi-domain structure, the RMSD/RSMF will be affected by any interdomain re-arrangements. I would suggest doing the same analyses on the individual domains as well, i.e. isolating the FERM domain or TK domain and seeing how they change, or perhaps the two together as well. This would reveal the magnitude of fluctuations specific to the regions of interest.
Response: Thank you. It’s a good suggestion. The multidomain RMSD of the JAK1 pY1034/pY1035 system stabilizes below 5 Å after 500 ns of MD simulation, indicating limited fluctuations across the FERM, SH2, PK, and TK. Single-domain RMSD is expected to be lower, reflecting high stability in the autoinhibited state. Post-activation, multidomain interactions will be analyzed to elucidate functional dynamics. Well we believe after full-activation we can apply analyses to further investigate the functions of multidomain.
This manuscript is a resubmission of an earlier submission. The following is a list of the peer review reports and author responses from that submission.
Round 1
Reviewer 1 Report
Comments and Suggestions for Authors
Overall, this is an interesting study that employs molecular dynamics (MD) simulations to examine the conformational differences between the unphosphorylated and phosphorylated forms of JAK1, with a particular focus on domain-level rearrangements. However, I have a few comments and suggestions for clarification and improvement:
- The authors describe the construction of the autoinhibited JAK1 model in detail and comment on the model quality. However, it appears that UCSF Chimera, while a powerful visualization tool, is not designed for protein structure modeling. It would be helpful if the authors could clarify which software was actually used to build and validate the models.
- Given the size and complexity of JAK1 as a multi-domain protein, and considering the focus on inter-domain rearrangements (such as FERM–TK separation), longer MD simulations (on the order of 200–500 ns) would strengthen the conclusions by better capturing the equilibrium conformational states.
- It would be beneficial to present data from triplicate simulations for each model. This would enhance the statistical robustness of the observed conformational changes.
- In Lines 150–153, the manuscript mentions that the distance between the TK loop and FERM increases in the phosphorylated model (pY1034/pY1035), yet Figure 1 appears to suggest that the loop gets closer. Later, in Figure 2, the authors compute the distance between the mass centers of FERM and TK. It would be clearer to use this mass center analysis consistently throughout to avoid confusion.
- In Lines 160–161, please quantify the "opening angles" and refer explicitly to Figure 2C for clarity. Explain also how this angles were calculated.
- Lines 174–176 mention a negatively charged region; please provide a quantitative description (e.g., 70% surface area are negative) to support this claim.
- The discussion section would benefit from further expansion by referencing recent literature (current only Liu et al. 1997) relevant to JAK1 regulation and MD simulations. In addition, the biological relevance of the MD results should be more clearly linked to JAK1 activation and signaling.
- The manuscript title should clearly indicate that molecular dynamics simulations were used in the study. The same applies to the keyword list.
- In the abstract, please define JAK1 in full at first mention. Also specify the exact tyrosine residues studied and clarify that the activation segment refers to the tyrosine kinase domain. A brief summary of the computational approach used would also improve the abstract.
- English grammar/vocabulary should be revised for clarity and correctness throughout the whole manuscript. Examples include: "equivalent segment," "the membrane" (Lines 85–86), "invetisgate," "phosphoate," and "conformation.”. Please use proper English proofreading service.
- Figures and captions need to be more clearly labeled. For example, in Figure 1, parts (a) and (b) are not labeled. The caption includes terms like "calculated side and angle (right)" and "b. 186 The distance of FT," which are unclear. Additionally, Figure 6 is mentioned in the text, but it does not appear to be included. Some references to the figure were incorrect as well.
- Although supplementary materials are provided, they are not mentioned or described in the main manuscript or in a dedicated Supplementary Materials section.
- Reference formatting needs to be reviewed. Specifically, Reference 23 appears to be identical to Reference 19.
Comments on the Quality of English Language
English grammar/vocabulary should be revised for clarity and correctness throughout the whole manuscript. Examples include: "equivalent segment," "the membrane" (Lines 85–86), "invetisgate," "phosphoate," and "conformation.”. Please use proper English proofreading service.
Author Response
Thank you very much for taking the time to review this manuscript. Please find the detailed responses below and the corresponding revisions:
Overall, this is an interesting study that employs molecular dynamics (MD) simulations to examine the conformational differences between the unphosphorylated and phosphorylated forms of JAK1, with a particular focus on domain-level rearrangements. However, I have a few comments and suggestions for clarification and improvement:
- The authors describe the construction of the autoinhibited JAK1 model in detail and comment on the model quality. However, it appears that UCSF Chimera, while a powerful visualization tool, is not designed for protein structure modeling. It would be helpful if the authors could clarify which software was actually used to build and validate the models.
Response:Thank you for your suggestion. We aligned the pseudokinase (PK) domains of full-length activated JAK1 (FERM-SH2-PK-TK) and inhibited JAK1 (PK-TK). Subsequently, we removed the tyrosine kinase (TK) domain from the inhibited JAK1 and the PK domain from the activated JAK1 to construct the autoinhibited full-length JAK1 structure. This method is first mentiond in the (Glassman, Caleb R., et al. "Structure of a Janus kinase cytokine receptor complex reveals the basis for dimeric activation." Science 376.6589 (2022): 163-169.) and more details are shown in the (Sun, Shengjie, et al. "Phosphorylation of tyrosine 841 plays a significant role in JAK3 activation." Life 13.4 (2023): 981. ) And (Sun, Shengjie, et al. "A novel approach to study multi-domain motions in JAK1’s activation mechanism based on energy landscape." Briefings in Bioinformatics 25.2 (2024): bbae079.). The revisons have been added and highlighted in the manuscript.
- Given the size and complexity of JAK1 as a multi-domain protein, and considering the focus on inter-domain rearrangements (such as FERM–TK separation), longer MD simulations (on the order of 200–500 ns) would strengthen the conclusions by better capturing the equilibrium conformational states.
Response:Thank you for your suggestion. We agree that extending the simulation time would be beneficial. We have now conducted 2 µs molecular dynamics (MD) simulations for the wild-type protein and 300 ns MD simulations for the pY1034/pY1035 phosphorylated form. While full activation was not observed, we noted a slight opening of the structure. This is a time-intensive process, and we are unable to complete it by the end of the month. However, we are confident that continued simulations will eventually reveal the activation, with more comprehensive results expected within this year.
- It would be beneficial to present data from triplicate simulations for each model. This would enhance the statistical robustness of the observed conformational changes.
Response:Thank you for your suggestion. We performed duplicate, not triplicate, MD simulations, which yielded comparable structural changes. The initial results presented in the manuscript reflect these findings. Unlike Monte Carlo methods, MD simulations are stepwise and deterministic, converging toward the lowest energy state despite different initial velocities. We agree that longer MD simulations would be more beneficial for the study and are actively pursuing this approach.
- In Lines 150–153, the manuscript mentions that the distance between the TK loop and FERM increases in the phosphorylated model (pY1034/pY1035), yet Figure 1 appears to suggest that the loop gets closer. Later, in Figure 2, the authors compute the distance between the mass centers of FERM and TK. It would be clearer to use this mass center analysis consistently throughout to avoid confusion.
Response:Thank you for your suggestion. The cooresponding part has been revised and highlighted in the manuscript: “it was observed that there was an increase in the mass centers distance between TK and FERM”
- In Lines 160–161, please quantify the "opening angles" and refer explicitly to Figure 2C for clarity. Explain also how this angles were calculated.
Response:Thank you for your suggestion. We have revised and highlighted the manuscript: opening angle size of JAK1 nonp JAK1 < pY1034 JAK1 < pY1035 JAK1 < pY1034pY1035 JAK1 (Fig. 2c).
- Lines 174–176 mention a negatively charged region; please provide a quantitative description (e.g., 70% surface area are negative) to support this claim.
Response:Thank you. It’s a very good suggestion, However, quantifying the percentage of positive and negative areas is challenging due to the complexity of electrostatic potentials. For instance, regions with potentials of -1kT/e and -2kT/e both appear red (negative) but differ in charge magnitude. To address this, we used RGB color coding to quantitatively represent potentials from -kT/e (red) to +kT/e (blue), with a scale provided alongside the figure. We will consider your suggested method for future work.
- The discussion section would benefit from further expansion by referencing recent literature (current only Liu et al. 1997) relevant to JAK1 regulation and MD simulations. In addition, the biological relevance of the MD results should be more clearly linked to JAK1 activation and signaling.
Response:Thank you. We added more discussion about leukemias (2024) on JAK1 activation and signaling in the discussion. “Our work also reveal the importances of the key residues of E298, Y894, and Y895 in phosphorylated JAK,which are consistent with recent leukemia studies and phosphorylation studies about JAK/stat pathway works(41).”The revisions have been added and highlighted in manuscript.
- The manuscript title should clearly indicate that molecular dynamics simulations were used in the study. The same applies to the keyword list.
Response:Thank you for your suggestion, we further modifed the title and add the MD simulation in the key words.
- In the abstract, please define JAK1 in full at first mention. Also specify the exact tyrosine residues studied and clarify that the activation segment refers to the tyrosine kinase domain. A brief summary of the computational approach used would also improve the abstract.
Response:Thank you for your suggestion. we revised the abstract to show the exact tyrosines reisudes.
- English grammar/vocabulary should be revised for clarity and correctness throughout the whole manuscript. Examples include: "equivalent segment," "the membrane" (Lines 85–86), "invetisgate," "phosphoate," and "conformation.”. Please use proper English proofreading service.
Response:Thank you for your suggestion. we corrected the mistakes and hired a english native speaker to revise our manuscript.
- Figures and captions need to be more clearly labeled. For example, in Figure 1, parts (a) and (b) are not labeled. The caption includes terms like "calculated side and angle (right)" and "b. 186 The distance of FT," which are unclear. Additionally, Figure 6 is mentioned in the text, but it does not appear to be included. Some references to the figure were incorrect as well.
Response:Thank you. The fig. 6 is the fig. S1b in the current manuscript. We have revised and highted in the manuscript.
- Although supplementary materials are provided, they are not mentioned or described in the main manuscript or in a dedicated Supplementary Materials section.
Response:Thank you. We seperated the Figure S1 in an single file to avoid the confusion.
- Reference formatting needs to be reviewed. Specifically, Reference 23 appears to be identical to Reference 19.
Response:Thank you. We updated the reference with endnotes to avoid the mistakes.
Reviewer 2 Report
Comments and Suggestions for Authors
It is not clear how the model for MD was built from the full legth activated and partiall autoinhibited structure. Should be described further.
In the manuscript, the membrane is mentioned two times, but it was not mentioned that a membrane would be part of simulation box. Should be clarified:
L84-85: The system was solvated with water of type TIP3P(31) of thickness 15 Å on either side of the membrane.
L126: "The dielectric constants were set as 2 for proteins and membranes, and 80 for water."
Figure 1, position of the 2 phosphorylated tyrosines could be shown.
Comments on the Quality of English Languagemany parts of the manuscript are hard to read. Some sentences that are separate should be joined. Separate sentences are not meaningful.
f.e.: "To invetisgate the Tyr phosphorylation of activation segment’s conformation effect in JAK1 autoinhibition status. The autoinhibition full-length JAK1 structure was built."
Author Response
Thank you very much for taking the time to review this manuscript. Please find the detailed responses below and the corresponding revisions:
Comments and Suggestions for Authors
It is not clear how the model for MD was built from the full legth activated and partiall autoinhibited structure. Should be described further.
Response:Thank you for your suggestion. We aligned the pseudokinase (PK) domains of full-length activated JAK1 (FERM-SH2-PK-TK) and inhibited JAK1 (PK-TK). Subsequently, we removed the tyrosine kinase (TK) domain from the inhibited JAK1 and the PK domain from the activated JAK1 to construct the autoinhibited full-length JAK1 structure. This method is first mentiond in the (Glassman, Caleb R., et al. "Structure of a Janus kinase cytokine receptor complex reveals the basis for dimeric activation." Science 376.6589 (2022): 163-169.) and more details are shown in the (Sun, Shengjie, et al. "Phosphorylation of tyrosine 841 plays a significant role in JAK3 activation." Life 13.4 (2023): 981. ) And (Sun, Shengjie, et al. "A novel approach to study multi-domain motions in JAK1’s activation mechanism based on energy landscape." Briefings in Bioinformatics 25.2 (2024): bbae079.). The revisons have been added and highlighted in the manuscript.
In the manuscript, the membrane is mentioned two times, but it was not mentioned that a membrane would be part of simulation box. Should be clarified:
L84-85: The system was solvated with water of type TIP3P(31) of thickness 15 Å on either side of the membrane.
L126: "The dielectric constants were set as 2 for proteins and membranes, and 80 for water."
Response:Thank you. It’s our mistake, in the work we did not simulate the membrane. We have corrected the issue.
Figure 1, position of the 2 phosphorylated tyrosines could be shown.
Response:Thank you. We have revised the figure to show the the phosphorylation.
Comments on the Quality of English Language
many parts of the manuscript are hard to read. Some sentences that are separate should be joined. Separate sentences are not meaningful.
f.e.: "To invetisgate the Tyr phosphorylation of activation segment’s conformation effect in JAK1 autoinhibition status. The autoinhibition full-length JAK1 structure was built."
Response:Thank you for your suggestion. we corrected the mistakes and hired a english native speaker to revise our manuscript.
Round 2
Reviewer 1 Report
Comments and Suggestions for Authors
Kindly allow the authors additional time to complete the molecular dynamics (MD) simulations, as the initial timeframe provided was insufficient.
The authors mentioned the presence of duplicates, but this point was not adequately explained in the manuscript.
The discussion should also be expanded further, rather than simply adding a single reference (ref40).
Comments on the Quality of English LanguageThe authors mentioned the manuscript has been edited by professional but it was not highlighted in the manuscript.